# Does Strategic Corporate Social Responsibility Drive Better Organizational Performance through Integration with a Public Sector Scorecard? Empirical Evidence in a Developing Country

**Pham Quang Huy [1,]*** and **Vu Kien Phuc [2]**

[1]  School of Accounting, University of Economics Ho Chi Minh City, 59C Nguyen Dinh Chieu Street, District 3, Ho Chi Minh City 70000, Vietnam

[2]  School of Accounting, University of Economics Ho Chi Minh City, Vinh Long Campus, 1B Nguyen Trung Truc Street, Vinh Long 85000, Vietnam; phucvk@ueh.edu.vn

*   Correspondence: pquanghuy@ueh.edu.vn; Tel.: +84-908-231260

**Abstract:**  This paper sets its sights on propounding a structural model to delve into the interrelationship between the impact of the integration of corporate social responsibility activities into the public sector scorecard management framework on the corporate social responsibility disclosure and enhancement of the organizational performance among public sector organizations. The conceptual framework in company with hypothesis framing were established after examining the related literature. Data were gathered from a sample of 723 respondents in public sector organizations in South Vietnam via convenience sampling method. Structural equation modeling was employed to validate the goodness of model fit and examine the hypotheses. These findings revealed that integration of corporate social responsibility activities into the public sector scorecard management framework was significantly and positively related to the corporate social responsibility disclosure and organizational performance. Additionally, it also asserted that corporate social responsibility disclosure was considerably associated in a positive manner with organizational performance. Thus, some detailed implications in connection with each causal relationship and several orientations were underlined to ameliorate the capacity of managing and measuring the organizational corporate social responsibility practices in a strategic manner.

**Keywords:** corporate social responsibility; corporate social responsibility disclosure; organizational performance; public sector scorecard

## 1. Introduction

Corporate social responsibility (CSR) practices have been broadly adopted throughout the organizational community in light of the value creation characteristic of CSR [1]. CSR's implementation can help an organization easily achieve the approval of local communities, good staff attraction and retention [2], far-reaching risk mitigation [3] and so on.

CSR has been widely applied in both developed and developing economies [4]. Surprisingly, the numerous scholars have largely paid attention to the developed countries while placing less concern on the developing economies [5].

In the developing economies, CSR represents the formal and informal ways in which a business creates a significant contribution to ameliorating the governance, social, ethical, labor and environmental conditions of the countries [6]. In addition, these are the regions where globalization, economic growth, investment and business activities are likely to cause the most significant social and environmental



impacts [7]. Furthermore, as stated by [8], the CSR report earned divergent attention from stakeholders in developed and developing countries. This is because the organizations in developing nations receive lower pressures in corporate social responsibility disclosure (CSRD). Asia has been the region most often covered in the literature in relation to CSR in developing nations [7]. Unfortunately, the research about CSR in this region within and among countries has still been sparse [9].

As a member of Asia, Vietnam first introduced CSR activities in 2000. The change in awareness of Vietnamese organizations was marked in 2010. Although a public sector organization (PSO) is more likely to perform socially responsible activities, these organizations are often found to concentrate on philanthropic activities rather than prioritizing environmental CSR initiatives to attain better organizational performance (OP). Admittedly, the biggest barriers and challenges to the implementation faced by PSO include the insufficient awareness of the concept of social responsibility; lack of financial and technical resources; and multiple sets of codes of conduct. Notably, the evaluation on the actual impact of CSR implementation was also another concern among many organizations [10]. However, PSOs have been lacking a clear, quantitative, consistent instrument for evaluating their goals. Moreover, according to government regulations, an alternation should be taken into consideration to meet the transparent and fair requirements when measuring the PSO's performance, especially in CSR's implementation.

In this regard, the management and measurement framework that has been proposed for this application is the public sector scorecard (PSS) due to the main advantages of outcome improvement economically; measurement development; public sector (PS) characteristics' conformity; emphasizing the expectations of users and stakeholders; re-designing phases and ameliorating service delivery; dealing with capability and organizational matters; and promoting a performance management culture based on improvement, innovation and learning [11]. Indeed, the PSS is considered as an effective framework for ensuring that strategy, processes and performance measures in relation to CSR practices are aligned with each other. In particular, these activities are also aligned with the demands and expectations of service users and stakeholders. Besides, PSS can create better conditions for the collaborative work between the leaders and staff and service users and other key stakeholders. It can also deal with risk management and organizational culture, and has the capability to assure that staff and processes are facilitated to attain the targets. In addition, PSS can give rise to the improvement of CSR practices and concentrate on creating desired outcomes, including value for money.

Building on these abovementioned analysis, this research generated a significant contribution to the literature in several aspects. Firstly, this paper placed an emphasis on the benefits and the likelihood of the integration between CSR and PSS (ICP) because the scrupulous CSR governance would lead to success via organizational structure enhancement, CSR performance accomplishment and sustainable development [12]. On the other hand, due to lacking independent verification, voluntary CSRD used to be criticized as being unreliable and reported under a selected manner [13]. Thus, the association between CSRD and ICP was handled to determine the effectiveness of ICP on the disclosure activities. Thirdly, the impact of ICP on PSOs' performance was also accentuated in terms of sustainable development. Although Asia has been broadly acknowledged as a region with numerous studies focusing on the CSR issues, the studies about CSR in the region within and among nations are still limited [9]. In addition, based on the assumption of [14], there have been limited theoretical and empirical investigations on CSR practices in Vietnam up to date. Thus, this research added new empirical evidence to ease the limitations of the amount of empirical research and the association between CSR and organizational outcomes [15]. These findings could help both practitioners and academicians attain a deep understanding on ICP through a general strategic management framework in connection with the CSR practices. In addition, these results also gain several useful insights into CSRD in PSO, as CSRD reporting has been shown to fail to attain homogeneous designs between countries [16]. Importantly, the proposed framework of the current research can be treated as an example for PSOs in other developing countries, especially the countries belonging to Asia, due to these following reasons. Firstly, as PSOs have been established with the primary role of serving

the public, they are required to deliver the highest levels of compliance with the law during their operations. Secondly, there have been several similarities in the characteristics of CSR practices [7]: the geo-economic development conditions and the same size of geo-economic flow between the countries in Asia [17]. To that end, this paper endeavors to illustrate how to adopt the theoretical model through addressing these research questions below.

**RQ1**. Does the ICP have a significant effect on the CSRD in PSO? How far does it influence?

**RQ2**. Does the ICP have a significant effect on the OP in PSO? How far does it influence?

**RQ3**. Does the CSRD have a significant effect on the OP in PSO? How far does it influence?

The remaining parts of this research are structured as follows. The review of the prior research is sketched out in Section 2. The next section institutes the theoretical background and develops hypotheses for the research. Subsequently, the methodology employed in empirical research is carefully elucidated in Section 3. Our main section in which the findings of the study are included is Section 4. Eventually, theoretical and managerial implications, and useful directions for future research based on the inherent limitations, are foregrounded in Section 5.

## 2. Overview of Prior Research

### 2.1. The Association between Corporate Social Responsibility and Management

Several scholars have recently become interested in the integration between CSR components and organizational management due to the perception on the improvement in corporate CSR target controlling [18]. In particular, [19] propounded the combination of financial and strategic control procedures to apply for managing the environment. Additionally, the sustainability ingredients were integrated into organizational strategy in terms of a management control system [20]. Analogously, [21] added a new finding by means of putting the management control system into CSR strategy management.

### 2.2. The Linkage between Corporate Social Responsibility and Organizational Performance

A large amount of research has been concentrated on exploring the impacts of CSR on the various facets of an organization [22], which provoked a variety of homogeneous results. Particularly, [23] indicated the causal link between corporate reputation, CSR and OP, whereas [24] argued that good CSR could lead to a good corporate reputation and gained the performance of the organization. Additionally, more involvement in the satisfaction of stakeholders was also another concern to ensure the OP, as with financial performance (FP) was also closely related to the image of the entities [25]. On the other hand, CSR implementation was supposed to lead to the satisfaction of the consumers regarding the quality of service and retain the highly qualified workers [26]. Thus, ultimately higher profits were promised. On the contrary, CSR has been argued to cause a negative effect on FP (i.e., [27]). In particular, there was less likelihood that FP could take place simultaneously with abounding CSR activities [28], as supplemental resources and capacity during the process of CSR process would cause high expenditures and lower profit [29]. As such, an inverse association between organizational CSR activities and FP has been ascribed to the entities with better CSR performance but failing to attain the financial facet [30]. Besides, in the investigation on the relationship between CSR, public service motivation (PSM) and organizational citizenship behavior (OCB) in PS, [31] proved that employee perceptions of both internal and external CSR influenced the development of a desire to serve the public in a positive manner. In addition, these findings indicated that PSM not only partially mediated the interconnection between internal CSR perceptions and employee OCB, but also fully mediated the association between external CSR perceptions and OCB.

### 2.3. The Relationship between Corporate Social Responsibility Disclosure and Organizational Performance

Researchers have recently placed their sharp-witted concern on the disclosures of CSRs [32]. However, the outcome of this subject has come to a conflict [33] in which the variety of measures of

CSRD, research methodologies and FP measures were ruminated to be the main causes [34]. While [35] manifested the evidence of the positive association between levels of CSRD in the annual statements and OP in relation to FP and corporate reputation, the CSRD-performance relationship was experimentally found to be less significant for practical purposes in the work of [36]. Furthermore, the association between CSRD and FP was also scrutinized through numerous empirical studies and was proved to be tight [37]. On the other hand, the converse results on the relationship between CSRD and FP were also highlighted in several studies (i.e., [38]). The neutral association between CSRD and FP also occurred in the findings of numerous scholars [39]. In the meanwhile, the evidence between CSRD and FP has not yet even been detected in several research (i.e., [40]).

## 3. Theoretical Background and Hypothesis Development

### 3.1. Theoretical Background

#### 3.1.1. Legitimacy Theory

*Legitimacy theory*. There has been a growing consensus among numerous scholars on putting legitimacy theory (LT) in place for explaining the driving force behind corporate social and environmental disclosures [41]. Therefore, this theory has been broadly applied in several studies related to the organizational environmental disclosures and has gradually assumed the central position in this type of research [42]. As stated by [43], the social contract is defined as multitudinous social expectations on the way which an entity should undertake its operations. LT is calculated based on the perception of a "social contract" existing between an entity and its operational surroundings [44]. According to [45], LT originated from the perception that an organization was supposed to consummate within the bounds and norms of with socially responsible behaviors. Simultaneously, organizational legitimacy leaned on the prolongation of reciprocal associations with its stakeholders [46], comprising implementing moral obligations to numerous stakeholders [47]. Building on the LT viewpoint, legitimacy and power have been consented to the organization by the society [48]. Hence, any particularly organizational behavior should be carefully investigated within its context and hunted for substitute driving forces [49], as these powers would have been lost if the organization had not utilized them in an appropriate manner. In doing so, the exploitable CSR activities could be effectively employed in almost all of organizations. On the other hand, as CSRD was considered to act as a critical mechanism to gain the impact of CSR on organizational reputation [50], it has been turned into a vital instrument for organizational management through integrating CSR activities into strategic risk management for the best result of CSR activities [51]. The more involvement in CSR activities, the more success in operation that this organization could reap [52].

*The application of Legitimacy theory in this study*. Within the association between organization and society, the CSR activities are persistently presented, investigated, identified and adjusted. The CSR reporting practices have become a primary instrument for organizational management [53] and maintaining legitimacy [43]. As such, if the chances of adverse shifts in community expectations become higher, the organization should make more effort regarding conducting the CSRD [43]. In other words, organizations attempt to legitimize their actions by engaging in CSR reporting to achieve acceptance from society. Integrating the CSR activities into the PSS framework should help by managing, measuring and improving the CSR activities. In doing so, the performance of CSR activities could be maximized. Accordingly, the PSO could determine which tactics and disclosure options would be available and suitable for managing legitimacy. Moreover, the PSO could take up numerous public disclosure strategies to gain the OP.

#### 3.1.2. Resource-Based Theory

*Resource-based theory*. Building on the resource-based viewpoint, the choice and accumulation of resources was considered as a function of internal decision making and external strategic

determinants [54]. Therefore, the resource-based theory could be used as an instrument for undertaking the analysis of social policy refinement [55]. As stated by [56], the organizational intangible resources consisted of technology, human capital and reputation. On the other hand, intangible resources could be comprised of the assets, capabilities, processes, attributes, information and knowledge managed by the organization [57]. Through combining intangible resources into strategic planning process, several researchers detached corporate social performance from OP and endeavored to bridge the former to organizational FP [58]. The CSR activities could give rise to a significant support for establishing and reinforcing a solid, sustainable, long-term reputation to enhance the competitive advantage [59]. Additionally, CSR practices could encourage the workforce, ameliorating productivity and facilitating improvement of performance [60].

*The application of resource-based theory in this study*. The considerable changes in the business environment have set a demand on organizational changes in terms of different resources and capabilities. An organization should have the capability to spread out its resources rather than only possessing unique resources. Thus, resource-side matters had to be addressed by the practicing strategists [61]. The resource-based view of the organization proposed that an organization should constitute its internal capabilities to match the conditions of the external environment. On the other hand, the theory inferred that the right combination of resources should be developed, progressively evaluated and managed for the specific OP intended. As such, PSOs have been advised to choose an appropriate framework to manage and measure the CSR activities based on organizational resources. In this case, PSS is considered as the most suitable framework for PSO [11]. Based on integrating the CSR practices into the PSS framework, CSR activities are aligned with each other. Additionally, these activities are also aligned with the expectations of service users and stakeholders. Besides, PSS can create better conditions for the collaborative work between the leaders and staff and service users and other key stakeholders. It can also deal with risk management and organization culture, and has the capability of assuring that staff and processes are enabled to attain certain targets. Furthermore, PSS can create improvements on CSR practices and concentrate on generating desired outcomes, including value for money. In doing so, it can make a significant contribution to the improvement in CSRD and enhancement in OP.

### 3.1.3. Corporate Social Responsibility and CSR Disclosure

*Corporate social responsibility*. CSR refers to a course of action in which the agreement to make a contribution for society and a cleaner environment was adopted in a voluntary manner by the organization [62]. It could be an approach for public image and reputational improvement through the activities that meet the needs of society [63]. The definition of CSR employed in the present research was insinuated by [64] which was mentioned as a consistent course of specific action and policies involved in stakeholders' satisfaction and the pivotal triple aspects of economic, social and environmental performance [64]. As propounded by [65], the two categories of CSR strategies for organizations to participate included the CSR governance in a serious and strict manner and CSR governance in a symbolic and opportunistic way. Particularly, a course of serious and rigorous operations was undertaken in terms of serious and strict CSR governance with the support of vital resources, which led to fruitful CSR outcomes [32]. Conversely, only corporate image or emergent matters were put in place for being addressed rather than dealing with essential resource allocation for a deep and strategic CSR program [66].

*Corporate social responsibility disclosure*. CSRD refers to the action of supplying financial and non-financial information presented in an annual statement or isolated social reports that was concerned with organizational interaction with its physical and social environment [67]. Besides, CSRD typically does not only consist of information on the physical environment, energy, human resources, products and community (presented in detail [68]), but also organizational operations, aspirations and public image in association with the environment, staff, consumer matters, energy utilization, corporate governance issues and so on [40].

### 3.1.4. Public Sector Scorecard

Being set up from the balanced scorecard adaption and extended for cultural and value establishment for public and voluntary sectors, PSS has been excogitated as a fructiferous framework with effective contributions to outcome improvement for service users and stakeholders without increasing overall expenditures [11]. PSS is administered through the three main junctures, including strategy mapping, service improvement and measurement and evaluation [11].

Strategy mapping is the process dealing with the association between outcome, process and capability components [69]. As such, a draft strategy mapping is built up after the series of interactive workshops on several matters regarding anticipated outcomes—strategic, service user, stakeholder, FP and capability outputs of the internal departments (i.e., senior managers, staff) and external components (i.e., service users and organizational stakeholders)—have been finished. A risk-management workshop transpiring through identification, lessening and eliminating of risks is subsequently used in the draft strategy map. Those efforts are finally complemented officially into the strategy map in terms of risk-management culture. Besides, disagreement on strategic drivers and diverge requirements and priorities should be balanced out [70] due to the diversity of targets and stakeholders [71].

The service improvement phase is set up with the aim of fostering the workshop participants to raise their voices and reveal the evidence or data for the generation of tools consisting of process maps, systems thinking and lean management to be complemented. Additionally, the next workshop confirms the capability outputs' attendance in the strategy with the purpose of bringing it to the attention of supporting staff, thereby creating a culture of improvement, innovation and learning rather than a quarrel culture.

In the measurement and evaluation step, the negotiation among workshop participants is held to find out the appropriate performance assessments for each constituents of the strategy map in which the potential evaluation approaches will be under investigation and selection. In addition, a comprehensive understanding on OP can be achieved through careful analysis and learning performance measures in the light of cause and effect definition opportunities and addressing issues of creation.

The cycle will end with utilizing the performance information for strategy map modification, the determination of further service renovations and the promotion of better performance evaluations. Nonetheless, in view of several changes and the association between performance measures and changing strategy appearances, the cycle will still go on [72].

### 3.1.5. Organizational Performance

As stated by [73], the success of an organization is reflected by the degree of its performance during operations. Owing to the multidimensional characteristics and multiple measurement approaches in existence, OP was argued to lead to numerous challenges in evaluation [74]. OP was established by the weighted combination of perceived and objective performance information which was proven to have a strong degree of convergence [75] and named such primary categories as the financial (i.e., objective) and nonfinancial (i.e., subjective) measures in a slightly different manner in the literature [76]. Notwithstanding the huge contribution to the examination on the relationship between CSR activities and OP based on the financial indicators [77], unfortunately, the financial indicators have no longer been sufficient for OP assessment, as the financial prosperity of an organization cannot be separated from social, environmental and governance activities [78]. Besides, the economic performance not been cared for properly, although the such vital components of the CSR concept as environmental, social and governance dimensions and the integration between CSR and environmental, social and governance implementation has been indicated to create the positive effect on the organizational value and performance [79]. Furthermore, there was an urgent call for environmental performance to be tacked on in OP evaluation in terms of sustainable development [80]. Apparently, a comprehensive framework, one which is multi-dimensional and qualitatively-based, has been in demand as a replacement [81].

### 3.2. Hypothesis Development

Owing to the same main target of serving the stakeholders, ICP in organizational CSR implementation management and measurement as follows.

*Service user and stakeholder involvement.* The effective public service design and provisions can have better outcomes if the participation of experience and knowledge provided by service users is highlighted [82]. Customers are considered to possess the power on account of the competition of discrepant dimensions among various entities [34]. As such, suitable methods for better customer treatment renovation are put in the CSRD [83]. Besides, owing to the vital role of stakeholders in organizational strategy, there has been growing attention on constructing and maintaining a close association, increasing the interaction between the organization and stakeholders to gather useful information [84] and creating a better chance for more innovative activity and achievement [85].

*Working across organizational boundaries.* Eliminating the boundaries between entities in operational processes is the most crucial affair to take into consideration, because the focus of service users is on the available service distribution through many organizations or departments [86] and the solution for Governmental achievement in terms of the main outcomes is the cooperation between numerous organizations [11]. Admittedly, the diversity in CSR practices in numerous developing regions based on their own perceptions about the CSR has led to a variety of CSRD designs. This, therefore, raises an urgent claim regarding PSS's application into a CSR program as it does not only allow the people coming from various departments or entities to put their attention toward common essential outcomes instead of narrower targets, but also helps people, regarding the measurement and assessment, to make precise evaluations of the outcomes, processes and capability components [11].

*Improvement and capability process.* The bulk of organizational resource consumption being from undertaking CSR activities has led to a barrier for organizations regarding investment in CSR programs [12]. In this regard, PSS is recommended to be adopted for this program due to the advantage of process improvement dealing with the different outcomes demanded, including financial outcomes in an overall performance management framework [11]. On the other hand, almost all CSR strategies are also covered with interior and exterior components [87]. In particular, the internal ingredients mentioned on the method were applied by the staff, while external components were related to the requirements and expectations of outside stakeholders [88]. Thus, human resource management (HRM) has been argued to make significant contributions to employee commitment boosting; the organizational commitment with CSR practices; and integrating creation between the CSR principles and HRM processes together with stakeholder alignment establishment [88]. Additionally, the leaders who possess the superior external expertise and knowhow would become the good observers for environmental protection regulations and place sufficient attention on corporate stakeholders [89]. As such, they may devote themselves to the social responsibility implementation and gaining information [90].

*Integrating risk management.* As stated by [86], the combination between identification and addressing the primary risks has been well-recognized among the effective performance entities. The comprehensive CSRD also leads to the accurate evaluation on the operation situation and organizational risk factors [89] and good reputation building and reputational risk avoidance [91]. On account of the effect of the customer on the organizational market risk, the treatment with organizational customers would be significantly improved and the risk of losing its share of market would be decreased considerably when the CSRD has been put in place [34]. Nevertheless, there are still numerous risks related to the operational process related to CSR practices which should be integrated into PSS management framework to attain a better performance of CSRD.

*Improvement, innovation and learning.* As asseverated by [92], CSR practices have positively related to an entity's performance in terms of the innovation process. As such, CSR activities have been supposed to assist with organizational innovation capacity which leads to the enhancement of distinction creation and competitive advantages [93] and the capacity for process and product

innovation [94]. In doing so, socially responsible organizations have been asserted to present the quantity of CSRD in an in depth and high-quality manner [95].

Based on the aforementioned information, the study hypothesis was formed.

**Hypothesis 1 (H1)**. *ICP has caused an impact on the CSRD in a significant and positive manner.*

CSR activities have been well-recognized to give rise to the enhancement in FP [96] and economic performance [97]. Indeed, CSR implementation was reported to provide a competitive advantage and help an organization to accomplish sustainable growth goals [98]. On the other hand, an organization which places more concern on CSR activities would increase its reputation, customer loyalty and employee satisfaction [99]. In terms of the association with OP in the stakeholder aspect, more involvement in CSR activities would positively impact OP due to the effective support in establishing a good relationship with stakeholders and better care for society [100]. In like manner, [101] also underlined that gaining better understanding among stakeholders of the measures implemented by the organization for their wellbeing would generate a rapidly increasing on OP. Numerous researchers have simultaneously placed an emphasis on the investment in stakeholder engagement and management due to its benefits of a positive image, high quality employee recruitment and employee retainment [102]. Thus, a research hypothesis was supposed as follows.

**Hypothesis 2 (H2)**. *ICP has caused an impact on the OP in a significant and positive manner.*

Owing to the vital role of information provided by the entities, CSRD was proved to bring a great deal of benefit for the organizations. Regarding the financial facet, CSRD implementation would lead to the prosperity in the economy [103]. In terms of the management aspect, CSRD was considered to have a significant influence on the effectiveness of interior decision-making [104] and exterior relationship management [105]. Importantly, proficiency in stakeholder management also helped the organization receive much useful support from their stakeholders [106] to accomplish better FP [107]. Besides, involvement in the CSRD was revealed to yield a competitive advantage and enhance a company's value [108]. Thus, a research hypothesis was considered as follows.

**Hypothesis 3 (H3)**. *CSRD has caused an impact on the OP in a significant and positive manner.*

The research model which was established on the LT and Resource-based theory to investigate the interrelationship between the ICP, CSRD and OP was depicted in Figure 1.

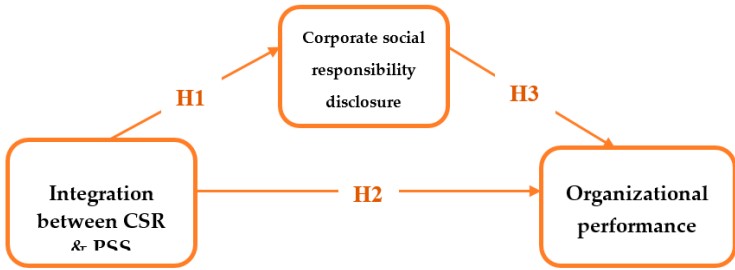

**Figure 1.** Proposed research model.

## 4. Methodology Design

### 4.1. Procedure and Item Generation

An empirical study was employed to verify the assumption model. Owing to its major role in the economic development of Vietnam, the southern region was selected to investigate the CSR practices among PSOs in the present study. Additionally, as PSOs have been established with the primary role of

serving the public, they are required to deliver the highest compliance levels with the law during their operations. Hence, almost all of the PSOs in Vietnam could take the results of this study as a reference. Moreover, due to the advantages of favorable environmental and economic conditions, this region has faciliated foreign investment much more than any other regions in the country. It is not surprisingly that this has been the region with the greatest developments in advanced management and modern technology adoption. Taken together, the findings on the research conducted in this region could serve as a reference for several developing countries in light of the similiarities in economic conditions.

In light of the adaption from English literature, all the scales applied in this study were used after translation and back-translation by a variety of bilingual experts. The questionnaire utilized in this research was set up with seven-point Likert scale ranging from 1 "completely disagree" to 7 "completely agree." Building on the above-mentioned research questions, the current study went through several procedures as follows.

Semi-structured interviews were employed in this study as they were considered to be suitable for gathering qualitative data from professionals [109]. Through employing semi-structured interviews, researchers could draw up a previous framework of themes to be investigate. Nonetheless, this type of interview could also help to emerge new ideas during the interview. The semi-structured interviews were done with several experts to consult their advice. The interviewees included four leaders of PSOs and four faculty members. Based on their suggestions, adjustments were made for several items which could not describe the current state of CSR practices in PSOs or were very hard to understand. Then, revisions with the experts were conducted again to create a complete questionnaire.

To improve sentence structure and layout of the instrument in relation to the PSOs in South Vietnam, a pilot test was performed. The pilot survey was undertaken with 100 participants randomly picked up from the target population. The Cronbach's $\alpha$ value was used to check the degree of internal consistency of each construct [110]. The Cronbach's $\alpha$ value of the pilot test were found to be above 0.7 [111], substantiating that the variables and dimensions of this research enclosed with acceptable reliabilities.

To examine the newly developed scale empirically, a cross-sectional study was employed for primary data collection through a questionnaire survey. As stated by [18], the accountants played an important role in measuring, disclosing and assuring all the organizational information, especially information about CSR. Hence, accounting staff were considered to participate in the generation, assurance, publication and analysis report on CSR regardless of the lack of formal stipulated structure [18]. As propsed by [112], the ideal sample size to estimate parameters for covariance-based SEM (CB-SEM) was 10:1. In the meanwhile, [113] suggested the optimal sample size which had an items to participants' ratio ranging from 1:4 to 1:10. The theoretical model contained 52 parameters and 3 indicators; thus, the number of 520 respondents was considered to meet the demand. Through the convenience sampling technique, these questionnaires were distributed from September 2019 to February 2020 and personally collected by the researchers. Unfortunately, some erroneous and incomplete questionnaires were excluded from the analysis. Finally, 723 complete responses were obtained, corresponding to a response rate of 87.84 % of the respondents. Hence, the sample was representative of the general population in the target region.

*4.2. Data Analysis*

As stated by [114], SEM was an effective statistical instrument for analyzing the interconnection between multiple variables by the measurement and structural models [115]. In the current research, SPSS version 25.0 was employed for evaluating the item-total correlations and exploratory factor analysis (EFA). In the meanwhile, AMOS version 25.0 was utilized for SEM. Maximum likelihood estimation method was applied to evaluate both measurement and structural model [116].

### 4.3. Measures and the Questionnaire

With the aim of exploring the impact of ICP on the CSRD and the OP in PSO, the measurement scales applied in this study were taken from the previous works in private sector due to several reasons.

Firstly, a thorough review of literature illustrated that far more concerns regarding CSR practices have been placed in the private sector [117]; it was not surprisingly that there have been numerous works devoted to finding out the appropriate measurement scales for CSR practices in this sector.

Secondly, the CSR practices in the developing countries have been most commonly related to philanthropy and charity [7]. In particular, these nations tend to conduct the corporate social investment in education, health, sport development and service communication. As stated by [118], PSO was underlighted to conduct more social and environmental commitments in comparison to that of private sector. PSOs have also undertaken numerous activities related to CSR pratices; namely, adhering to strict regulations, helping the poor, operating in a manner in line with the philanthropic and charitable expectations of society, contributing toward bettering the local communities and so on.

Thirdly, under budgetary pressures and program effectiveness enhancement, public leaders typically seek a new approach to managingg the organization through adopting practices from the other sectors [119]. In addition, owing to the demand of adherence to strict regulations, PSOs have been made to experience the structural and procedural changes applied in the private sector [120].

Taken together, the measurement scales employed in this study were inherited from the previous works investigated in private sector. Nevertheless, the measurement scales used the revisions of experts and the pilot test to achieve appropriateness with the context of this study. In a nutshell, the measure scales employed in the current research were set up as follows.

### 4.3.1. The Integration between Corporate Social Responsibility and Public Sector Scorecard

There has been a growing concern in measuring the actual impact of CSR implementation [10]. The most critical demands of an effective performance management system included quality management, service redesign and performance measurement [69]. Through integrating with the PSS, the organizational strategies, processes and performance measures on CSR practices should accord with the outcomes related to service users and other key stakeholders.

As stated by [121], concentrating on the outcomes could guide the organization towards the true goals and enhance the accountability. Additionally, concentrating on the outcomes and being able to evaluate them lets the organization measure what organizational activities are actually being achieved [122]. As such, the integration of PSS into the organizational operation could bring several advantages. According to [11], the PSS project was begun with identifying the outcomes of PSO, its service users and other stakeholders and value for money which helped the PSO concentrate on these outcomes. Based on those things, the measurement scales for the outcomes of CSR activities when these activities were integrated with PSS were established as follows.

*Key performance outcomes*. The key performance outcomes were defined as the main performance outcomes demanded by the relevant organization [69]. Thus, the items for measuring the key performance outcomes in PSO in the research were developed from the works of [123,124].

*Service user/stakeholder*. The service user/stakeholder was referred to the outcomes relating to the service users and other key stakeholders. Therefore, the measurement scales of service user/stakeholder in this study were designed from those propounded by [123–125].

*Financial*. The financial was identified as the accomplishment of value for money or decreasing overall cost. The criteria applied to measure the financial were taken as reference from the contributions of [123,124].

It was distinguished from planned service and policies because it dealt with the actual experience of users and stakeholders [11]. The organization would try to determine how the processes could be improved to generate better performance.

*Service delivery*. The items for measuring the Service delivery in this study were modified from the works of [123,124].

The capability in PSS framework focused on what could be conducted to assure that redesigned processes performed smoothly. This might relate to extra resources to enable the organization to accomplish the required outcomes [69]. These extra resources typically comprised innovation and learning; effective leadership; and people, partnerships and resources. To that end, the measurement scales for these components were set up as follows.

*Leadership*. The significance of leadership towards success in CSR practices has been underlined ([126–129]). The criteria applied to measure the leadership in this study were adjusted from the contributions of [130].

*People, partnerships and resources*. Because CSR implementation also raise the demand on by generous amount (of organizational resources; [12]), the organization had to allocate resources in an appropriate manner [131]. Thus, in this study, the measurement scales of the People, partnerships and resources were modified from the works of [123,124].

*Innovation and learning*. As proposed by [132,133] CSR could serve as a source of innovation and competitive advantage, analyzing and learning from performance measures could offer deep understandings on how effectively the organizations performed [69]. Hence, the measurement scale of Innovation and learning were adjusted from the works of [26,125,134].

### 4.3.2. Corporate Social Responsibility Disclosure

Apart from the above-mentioned reasons for the selection of measurement scales in this research, our choice of modifying the measurement scales established by [135] was due to the correspondence between the two research contexts. To put it simply, the context of this study was the developing country and the research undertaken by [135] was also in the developing region. In doing so, CSRD was made up by the five primary components including Community Welfare, Contribution to Education, Environmental and Energy Importance, Services, Customers and Stakeholders and Workforce. Accordingly, Community Welfare included three sub-scale items was modified from the study of [40,135–137], and [33]. In the meanwhile, the rest of the measurement scales was adapted from the findings of [135].

### 4.3.3. Organizational Performance

The organizational financial prosperity should be fulfilled with the appearance of social, environmental, and governance activities [78]. Thus, the measurement scales for ORG employed in this paper were established by the four key elements namely Economic performance, Environment performance, Human performance, Governance performance.

Economic performance connected to the economic condition which focused on the economic indicators instead of financial indicators presented on the annual statement [138]. As such, Economic performance was adapted from the works of [139].

Human performance signified to the association between the organization and its labor force [140]. Thus, Human performance was aligned from the studies implemented by [141].

Environmental performance described the endeavor which organizations utilize to insulate nature [142]. Environment performance was adjusted from the works of [125,134,143,144].

Governance performance implied for the systems and processes related to sheltering the organizational orientation, control and accountability [145]. Additionally, board composition and board behavior [146] and satisfying stakeholders [147] were supposed to be the main targets of organizational governance. Hence, Governance performance was taken from the outcomes of [148].

## 5. Result Analysis and Discussion

### 5.1. Demographic Characteristics

The demographic profile of the 723 respondents was covered with their gender, age, qualification and working experience. In terms of the gender, females constituted 75.10 per cent of the respondents

while only 24.9 per cent of males were devoted to the main sample. With regard to the age, 196 respondents (7.47 per cent) belonged to "above 45" group, 328 respondents (45.37 per cent) were the "35–45" group, 145 respondents (20.06 per cent) were "25–35" group and the remaining 54 respondents (7.47 per cent) were classified as "below 5." The work experience ranged from below 5 years (7.47 per cent) to 5–10 years (20.75 per cent), 10–15 years (49.41 per cent) and more than 15 years (23.62 per cent). Moreover, respondents having an undergraduate background accounted for 94.47 per cent, whereas respondents having a postgraduate degree took up a tiny minority (5.53 percent) of the target population.

*5.2. Assessment of Convergent Validity*

The reliability analysis of the scale was firstly carried out through evaluating the Cronbach's $\alpha$. Hence, value of the Cronbach's $\alpha$ was recommended at 0.7 or more to demonstrate the trustworthiness of the scale [149]. Given that convergent validity illustrated the extent to which the scale correlated positively with other measures of the same constructs [150], factor loadings, composite reliability (CR) and average variance extracted (AVE) were employed in this study for convergent validity measurement [151]. Thus, standardized factor loadings were suggested to exceed the value of 0.6 [152]. Besides, CR was requested to be over the cutoff value of 0.82 [153]. The acceptable level of AVE was expected to be above 0.5 [154]. The results depicted in Table 1 indicated that the model obtained good convergent validity.

**Table 1.** Results summary for the measurement model.

| Model Construct | Items | Factor Loadings Ranges | AVE | Cronbach's Alpha | Composite Reliability | Discriminant Validity | Source |
|---|---|---|---|---|---|---|---|
| **Integration Between Csr and Pss** | | | | | | | |
| Key performance outcome | 4 | 0.748–0.854 | 0.614 | 0.861 | 0.864 | Yes | |
| Financial | 2 | 0.839–0.854 | 0.719 | 0.835 | 0.837 | Yes | [123,124] |
| Service delivery | 4 | 0.722–0.805 | 0.595 | 0.852 | 0.854 | Yes | |
| People, partnerships and resources | 3 | 0.823–0.858 | 0.698 | 0.871 | 0.874 | Yes | |
| Service user/stakeholder | 3 | 0.817–0.877 | 0.714 | 0.882 | 0.882 | Yes | [123–125] |
| Leadership | 3 | 0.806–0.889 | 0.710 | 0.878 | 0.880 | Yes | [130] |
| Innovation and Learning | 3 | 0.826–0.888 | 0.734 | 0.892 | 0.892 | Yes | [26,125,134] |
| **Corporate Social Responsibility Disclosures** | | | | | | | |
| Community Welfare | 3 | 0.795–0.888 | 0.696 | 0.872 | 0.873 | Yes | [33,40,135–137] |
| Contribution to Education | 2 | 0.832–0.910 | 0.759 | 0.862 | 0.863 | Yes | |
| Environmental and Energy Importance | 3 | 0.814–0.874 | 0.723 | 0.884 | 0.887 | Yes | [135] |
| Services, Customers and Stakeholders | 3 | 0.832–0.883 | 0.745 | 0.893 | 0.898 | Yes | |
| Workforce | 3 | 0.802–0.889 | 0.713 | 0.877 | 0.881 | Yes | |
| **Organizational Performance** | | | | | | | |
| Economic performance | 4 | 0.707–0.821 | 0.622 | 0.866 | 0.868 | Yes | [139] |
| Environment performance | 3 | 0.790–0.835 | 0.669 | 0.858 | 0.858 | Yes | [125,134,143,144] |
| Human performance | 5 | 0.705–0.804 | 0.562 | 0.864 | 0.865 | Yes | [141] |
| Governance performance | 4 | 0.709–0.861 | 0.607 | 0.859 | 0.860 | Yes | [148] |

## 5.3. Assessment of Discriminant Validity

The discriminant validity was considered as the extent to which measures of a given construct distinguished from measures of other constructs in the same model [112]. The AVE could be wielded to ascertain the discriminant validity [154]. Accordingly, when the AVE of each of the latent constructs was higher than the highest squared correlation compared with any other latent variable, discriminant validity of the construct level was set up [155]. The Table 2 displays that the square root of the AVE values were well above the correlation values; thus discriminant validity was achieved.

**Table 2.** Results of discriminant validity.

| | HP | EP | KPO | GP | SD | SCS | IL | SUS | EEI | WORK | LEAD | CW | PPR | ENP | CE | FINA |
|---|---|---|---|---|---|---|---|---|---|---|---|---|---|---|---|---|
| **HP** | 1 | | | | | | | | | | | | | | | |
| **EP** | 0.034 | 1 | | | | | | | | | | | | | | |
| **KPO** | 0.243 | 0.090 | 1 | | | | | | | | | | | | | |
| **GP** | 0.128 | 0.171 | 0.093 | 1 | | | | | | | | | | | | |
| **SD** | 0.061 | 0.065 | 0.064 | 0.186 | 1 | | | | | | | | | | | |
| **SCS** | 0.097 | 0.071 | 0.073 | −0.005 | 0.012 | 1 | | | | | | | | | | |
| **IL** | 0.146 | −0.010 | 0.040 | 0.076 | 0.133 | 0.065 | 1 | | | | | | | | | |
| **SUS** | 0.014 | −0.026 | 0.156 | −0.011 | 0.037 | 0.078 | 0.025 | 1 | | | | | | | | |
| **EEI** | 0.318 | 0.145 | 0.121 | 0.108 | 0.037 | 0.150 | 0.078 | −0.001 | 1 | | | | | | | |
| **WORK** | 0.146 | 0.191 | 0.112 | 0.047 | 0.030 | 0.228 | 0.124 | 0.107 | 0.013 | 1 | | | | | | |
| **LEAD** | −0.023 | −0.017 | 0.044 | 0.006 | 0.172 | 0.027 | 0.213 | 0.160 | 0.037 | 0.100 | 1 | | | | | |
| **CW** | 0.123 | 0.051 | 0.106 | 0.070 | 0.063 | 0.136 | −0.024 | 0.035 | 0.209 | 0.110 | 0.143 | 1 | | | | |
| **PPR** | 0.158 | 0.038 | 0.099 | 0.008 | 0.229 | 0.016 | 0.174 | −0.160 | 0.056 | 0.123 | 0.168 | 0.074 | 1 | | | |
| **ENP** | 0.121 | 0.212 | 0.131 | 0.147 | 0.049 | 0.043 | 0.065 | 0.062 | 0.067 | −0.043 | 0.004 | 0.089 | 0.003 | 1 | | |
| **CE** | 0.062 | 0.019 | 0.119 | 0.071 | 0.027 | 0.095 | 0.072 | 0.066 | 0.159 | 0.126 | 0.054 | 0.233 | −0.017 | 0.067 | 1 | |
| **FINA** | 0.231 | 0.035 | 0.195 | 0.074 | −0.023 | 0.047 | 0.000 | 0.194 | 0.036 | 0.190 | −0.071 | 0.077 | 0.008 | 0.024 | 0.017 | 1 |

HP = human performance; EP = economic performance; KPO = key performance outcome; GP = governance performance; SD = service delivery; SCS = services, customers and stakeholders; IL = innovation and learning; SUS = service user/stakeholder; WORK = workforce; EEI = environmental and energy Importance; LEAD = leadership; PPR = people, partnerships and resources; CW = community welfare; ENP = environment performance; CE = contribution to education; FINA = financial.

## 5.4. Assessment of Overall Model Fit

The generally lowest values suggested for GFI, TLI, AGFI and CFI were 0.90 and the ratio of $\chi^2$/df was proposed to be below 3.0 [156]. On the other hand, the value of GFI was also reported to be under 0.95 in several research namely the GFI index ranging from 0.774 to 0.923 [157]. The results in the Table 3 exposed that the measurement model and structural model met the goodness of fit requirements in the present context.

**Table 3.** Results of measurement and structural model analysis.

| The Goodness of Fit Measures | CMIN/DF | GFI | CFI | TLI | RMSEA |
|---|---|---|---|---|---|
| Recommended value | ≤3 | ≥0.9 | ≥0.9 | ≥0.9 | ≤0.08 |
| Measurement Model | 1.810 | 0.903 | 0.953 | 0.964 | 0.033 |
| Structural Model | 1.946 | 0.887 | 0.940 | 0.937 | 0.036 |

## 5.5. Hypothesis Verification

### 5.5.1. Direct Effect

In order to test the research hypotheses, this study estimated the path coefficients of the research statistical structural model which revealed several noticeable results highlighted in Table 4 as follows.

**Table 4.** Structural coefficients (β) of the proposed model.

| Hypothesis | Relationship | | | Estimate | S.E. | C.R. | P | Inference |
|---|---|---|---|---|---|---|---|---|
| **Hypothesis 1 (H1)** | CSRD | ← | ICP | 0.544 | 0.137 | 3.960 | 0.000 | Supported |
| **Hypothesis 2 (H2)** | OP | ← | ICP | 0.291 | 0.117 | 2.482 | 0.013 | Supported |
| **Hypothesis 3 (H3)** | OP | ← | CSRD | 0.272 | 0.112 | 2.427 | 0.015 | Supported |

The outputs illustrated that the positive effect of ICP (β = 0.544) was significant at the 95% confidence level, hence offering support for Hypothesis 1 (H1), which conjectured that ICP had a positive influence on the CSRD. This indicated that in the current study the effect of ICP on CSRD was significant. In other words, PSO could succeed in gaining the efficiency and effectiveness in disclosing CSR issues through putting the ICP into action.

The research results propped up Hypothesis 2 (H2) which was developed to investigate the effect of ICP on the ORG (β = 0.291). This hinted the positive effect of ICP on OP. In other words, the ICP would facilitate the PSO to enhance the overall performance. Accordingly, undertaking CSR practices under the PSS framework in a strategic manner could generate significant support for attaining higher performance in a variety of ways; namely, in economic, human, environmental and governance facets.

In order to examine Hypothesis 3 (H3), the impact of CSRD on OP was measured. The results showed that the effect of CSRD on OP was significant at the 95% confidence level (β = 0.272). Since gaining better understanding among stakeholders on the measures implemented by the organization for their wellbeing would generate a rapid increasing on OP [101]. Hence, the higher degree in the CSRD that could be achieved by the PSO, the better performance the PSO could reap.

### 5.5.2. Indirect Effect

The results showed the presence of the positive indirect effect of ICP on OP through CSRD was significant at the 95% confidence level. Therefore, if the level of CSRD gets higher, the mediating effect between ICP and OP is significant (β = 0.506), supporting the positive indirect effect of ICP on OP in PSO. By controlling the mediators, the direct effect of ICP on OP was significant but weaker (β = 0.257), indicating full mediation. To put it simply, PSOs could enhance their overall performance in such aspects as economic, human, environmental and governance facets when the disclosure practices were taken into consideration instead of concentrating only on performing the ICP.

## 6. Concluding Remarks

### 6.1. Discussion and Implication

From the academic standpoint, the current research has augmented on the studies related to the strategic CSR management toward the sustainable development. Although the combination between CSR and management has been not a new concept and substantiated to bring several certain effectiveness to the organizations, it has been widely applied in the private sector, which has many differences in characteristics compared to PSOs. In this regard, the research has deepened the insights on the likelihood and the advantages of integration between CSR practices into a management framework which is well-acknowledged to be best suited to the characteristics of a PSO. In light of the same target, stakeholder, PSS has been proved to be in appropriate to integrate the CSR practices in PSO. Importantly, this potential integration has empirically validated in this study to uplift the CSRD in the developing countries, which was regarded as the pressing concern in these regions as CSRD has not yet been compulsory regulation [158] and the demand on CSR programs due to the inherent social provisions and governance gaps [159]. On the other hand, the significant and positive impact that ICP effectuated on OP has invigorated the role of PSS adoption in relation to sustainable development. This framework is deliberated for workshop established approach in which the outcome definition has been implemented through raising spirits of the organizational internal and external components to participate to restructure and originate the new process to accomplish these outcomes and find out the effective solution for the capability and related matters to attain the outcome. As such, it can be taken as a reference for addressing the issue of lacking similarities in CSR practice and reporting as the boundary between regions and organization has eliminated. What is more, the outcomes mentioned in this model mainly come to grips with the target which is under the highest satisfaction from almost all of the users and stakeholders who played the most important role in the existence of the organization. Additionally, the financial outcomes are also involved to boost the financial situation

of the organization, and simultaneously, generate a better solution to the prior expostulation based on the fact that the entities with better performance in CSR practices would suffer from the poor FP [30]. Importantly, service delivery chiefly revolve about actual experiences of users and stakeholders will lead to the higher opportunities to achieve loyalty of the users and the approval of the stakeholders. Besides, the capability in this framework also buttons down on the essential support for the employees (i.e., motivation and training) and processes (i.e., innovation and learning, leadership) in fulfilling the outcomes and outputs.

Additionally, the findings of this research have put accent on CSRD with the roles both on the independent and mediate variable. In terms of the role as an independent variable, the findings on the interconnection between CSRD and OP have elucidated and reinforced the works of several prior researchers (i.e., [35,160]). In the meanwhile, this study has given out a converse result to some prior works (i.e., [161]) in terms of the mediating role of CSRD.

Furthermore, the performance of PSO has been concentrated on the comprehensive aspects in terms of sustainable development, namely, the economic performance; environment performance, human resource performance and governance performance which highlighted the consistence of the ICP can be best suited for managing and measuring the CSR in a strategic manner toward the sustainable development in PSO in the developing regions.

In respect to the practical implication, the leaders are supposed to intensify the perception of CSR practices as a reasonable investment rather than an obligation to conduct. Besides, the result which accentuated the contribution of PSS into managing and measuring the CSR program in PSO in terms of the strategic manner toward the sustainable development also raises a demand on the consideration for adoption in PSO. As such, with the exception from the certain support of financial resources, the departmental communication also play an important role in facilitate for the introduction and application of PSS as the employees will be likely to adopt in effective manner with the comprehensive perception on the meaning and value of PSS and CSR practices. On the other hand, the findings on the mediating of CSRD in the ICP-OP linkage have raised an urgent demand for policymakers for a general regulation about CSRD towards sustainable development in PSO because environmental outcomes can only be achieved through policy instruments which based on the combination of laws, regulatory approaches and market signals to cause a positively significant impact on operations and behaviors [162] among PSOs.

### 6.2. Limitations and Further Research

Unfortunately, the current research still suffered from several limitations. The restriction in terms of small sample size and target population was considered as the primary barrier in the generalization of empirical outcomes. Nevertheless, this situation will probably be eliminated when such a manifold population has been targeted in further studies. Owing to the cross-sectional data collection, the findings of the present research were prevented from creating any strong causal requirements in relation to these effects [163]. Thus, future researchers should take the longitudinal designs into consideration to allow the research framework to be modeled over time. Importantly, the replication of this type of research in a variety of areas has been in demand due to desires to underline much more interesting findings on the relationships of the proposed model in detail; the introduction of PSS and the number of issues regarding CSR are still too much for PSOs to handle, especially in the developing economies. Besides, the secondary data related to CSR issues are also recommended for the future work to eradicate the inherent limitations in this study. Eventually, we request the involvement of several procedures in relation to the direct-indirect for the more accurate affirmation.

**Author Contributions:** P.Q.H. established the conception and design of the research. V.K.P. conducted the data acquisition, analysis and interpretation. V.K.P. prepared the drafted paper. P.Q.H. provided vital revision of the paper. P.Q.H. offered final approval of the version to publish. P.Q.H. was accountable for the accuracy or integrity of any part of the work. All authors have read and agreed to the published version of the manuscript.

**Funding:** This research was funded by University of Economics Ho Chi Minh city, Vietnam

**Conflicts of Interest:** The authors declare no conflict of interest.

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
