# Peer review of "Does Strategic Corporate Social Responsibility Drive Better Organizational Performance through Integration with a Public Sector Scorecard? Empirical Evidence in a Developing Country"

_processes, doi:10.3390/pr8050596_

Round 1
Reviewer 1 Report
Dear author(s),
I congratulate you because the topic is very interesting and the statistic method is adjusted to analyze it. However, I must suggest on several topics:
- I consider that you need to rewrite the paper most clealy. In general terms, the manuscript is like a "word search", and the sentence about the main is a light example.
- In relation to the Theoretical background, I can´t understand the relationship between this study and Stakeholder theory. I think that there are current theories which seem to support the hypothesis development. I suggest looking up studies about the Common agent theory and including a new theorical background.
- I don´t undertand several definitions of variables because the used terms of dimenssions are too much ambiguous and an not much description about them. I recommend that you provide more explanation about your approach.
I hope that these recommendations are useful for you in improving the paper and that your study will be successful.
Author Response
|
No |
Comments and Suggestions |
Response of the authors |
|
1 |
Introduction |
The introduction has been rewritten |
|
|
Research design |
The research design has been rewritten |
|
|
Research methods |
The research method has been rewritten |
|
|
The words applied in the paper |
There has been numerous sentences being rewritten |
|
|
The Theoretical background |
The new theories have been added |
|
|
The definitions of variables |
Several definitions of variables have been rewritten with clear explanation |
|
|
|
|

Reviewer 2 Report
Thank you very much for the interesting research and the useful insight into a relatively unexplored topic. I´d like to make some comments that I hope will help to improve your research.
The use of so many acronyms (specially in the introductory part) can be somehow confusing, I´d recommend to substitute some of them with more explanatory texts. I´d suggest to put a list of acronyms at the beginning/end of the text to ease the reading. The abstract gets confusing since the acronyms are not described.
The figure 2 it´s not clearly visible, which should be improved. The redaction of the hypotheses could be improved, I am not sure that the verb "effectuated" is the best fit.
About the results, I´d suggest to explain a little more the point 5.5 about Hypothesis verification. In there you just put the results but do not really tell what they mean in practical terms (although they are covered in the next section, an extra paragraph would be useful).
And last but not least, some of the bibliographic references could be updated to cover more recent studies in the field like https://www.researchgate.net/profile/Sudhir_Saha/publication/328119631_Corporate_Social_Responsibility_Public_Service_Motivation_and_Organizational_Citizenship_Behavior_in_the_Public_Sector/links/5d88dcbc458515cbd1b89cc4/Corporate-Social-Responsibility-Public-Service-Motivation-and-Organizational-Citizenship-Behavior-in-the-Public-Sector.pdf
Author Response
|
No |
Comments and Suggestions |
Response of the authors |
|
2 |
The introduction |
The introduction has been rewritten |
|
|
The results of the paper |
The result analysis has been rewritten |
|
|
The abstract and introduction suffered from many acronyms |
The abstract has been rewritten |
|
|
The figure 2 and the verb "effectuated" |
The figure 2 has been changed The verb "effectuated" has been alternated |
|
|
Hypothesis verification |
The Hypothesis verification has been modified |
|
|
The bibliographic references |
The bibliographic references have been added |
|
|
|
|

Reviewer 3 Report
The paper investigate the relationship between the impact of the integrating CSR activities into public sector scorecard (PSS) management on the CSR disclosure to improve the organizational performance (ORG) in public sector organization (PSO).
The topic is very interesting and the paper is well written.
The some minor aspects to improve:
The second part of the title can be change with "Empirical Evidence in Vietnam".
In abstract I recommend do not use the abbreviation for INTE, PSS, ORG, PSO, etc.
Please mention in the paper how relevant are the sample from your research and how can be extrapolated the results to entire country (Vietnam) and to other developing countries.
Author Response
|
No |
Comments and Suggestions |
Response of the authors |
|
|
|
|
|
3 |
The introduction |
The introduction has been rewritten |
|
|
The result of the paper |
The result analysis has been rewritten |
|
|
The second part of the title can be change with "Empirical Evidence in Vietnam". |
The second part of the title has been changed |
|
|
The abbreviation for INTE, PSS, ORG, PSO, etc in the abstract |
The abstract has been changed |
|
|
Please mention in the paper how relevant are the sample from your research and how can be extrapolated the results to entire country (Vietnam) and to other developing countries. |
An explanation has been performed to prove that the sample of this research can be extrapolated for the entire country and other developing countries |

Round 2
Reviewer 1 Report
REVIEW of manuscript in processes-769259, with tittle DOES STRATEGIC CORPORATE SOCIAL RESPONSIBILITY DRIVE BETTER ORGANIZATIONAL PERFORMANCE THROUGH INTEGRATING WITH PUBLIC SECTOR SCORECARD? EMPIRICAL EVIDENCE IN DEVELOPING COUNTRY
Dear author(s)
In my opinion, this research provides some useful results and a good methodological strategy. However, the general main is not an original proposal because there are a lot of previous studies which use similar topic. Therefore, I recommend accept this paper but whit major revisions.
Then I propose another questions which can work out well for publishing:
- In relation to the Introduction, a description of actual management system in South Vietnamese public sector is lacking and the impact of Scorecard model in this geographical and sectorial context.
- In relation to the theoretical background, the author(s) focus on describing two theoretical approach, but don´t fall to respond clearly the main requests; why should public organizations “embed” corporate social responsibility criterion into his management system? How should they embed these criterion in such management?
- Regarding of variables, it is required to add more explanation about the items that author(s) deduced of previous studies. These are making reference to enterprise context and the context of this research is the public administration. In this sense, I assume that author(s) needed to remedy some of them.
- According to the writing style, I have to mention that the reading this paper is difficult for several reasons. First, acronyms and abbreviations in the text are unreasonable. Moreover, there are grammatical structures which are too long. As consequence, I suggest a review of a good editing.
Furthermore, I am convinced that a restructuring of the Introduction and theoretical background is needed because I notice the absence of logical-deductive when I read the manuscript.
I hope that you can use these suggestions to improve your paper
Author Response
LETTER OF REVISION PAPER BASED ON REVIEWER 1 – ROUND 2
In relation to the Introduction, a description of actual management system in South Vietnamese public sector is lacking and the impact of Scorecard model in this geographical and sectorial context.
The introduction (line 33-106) has been restructured based on the recommendations of reviewer. As such, the introduction has been presented as follows:
The benefit of CSR activities has been outlined. Nextly, the researchers have place the concern on the state of CSR activities in developing countries and Vietnam to seek for the gap of this type of research. The actual management system in South Vietnamese public sector has been highlighted through the implementation and management of CSR practices (line 59-62). Nextly, the introduction and the impact of Public sector scorecard on the CSR practices has been clearly depicted (line 63-76). Besides, the researchers also rewrote the contribution of the current research based on the mentioned adjustments.
In relation to the theoretical background, the author(s) focus on describing two theoretical approach, but don´t fall to respond clearly the main requests; why should public organizations “embed” corporate social responsibility criterion into his management system? How should they embed these criterion in such management?
Regarding to the recommendation on giving explanation on “why should public organizations “embed” corporate social responsibility criterion into his management system”, a paragraph has been added (line 186-196) to generate a clearly explanation on why PSO should apply CSR into management system.
In term of the recommendation on giving explanation on “How should they embed these criterion in such management”, a paragraph has been added (line 211-228) to comprehensively present how PSO should apply these criteria in the Public sector scorecard.
Regarding of variables, it is required to add more explanation about the items that author(s) deduced of previous studies. These are making reference to enterprise context and the context of this research is the public administration. In this sense, I assume that author(s) needed to remedy some of them.
In response to these recommendations, the researchers have been conducted the restructure for the methodology in this research. Accordingly, Procedure and item generation (line 414-459) has been rewritten to attain the clearly presentation on how the research were performed. Data analysis (line 461-466) has also been rewritten to demonstrate how the data were analyzed. Notably, the Measures and the questionnaire has been modified based on the request of the reviewer. As such, the clear explanation (line 468-491) on why the measurement scale employed in private sector could be used in this context (Public sector). Additionally, each of variable in the theoretical model have been explained intensively on several features such as the definition, the measurement scale applied for each variable. As such, the integration between corporate social responsibility and public sector scorecard (line 493-541) has been started with the explanation on the benefit of this integration. Nextly, each item has been presented with its definition and how to measure. Similarly, the Corporate social responsibility disclosure (line 543-522) and Organizational performance (line 558-571) have been clearly depicted.
According to the writing style, I have to mention that the reading this paper is difficult for several reasons. First, acronyms and abbreviations in the text are unreasonable. Moreover, there are grammatical structures which are too long. As consequence, I suggest a review of a good editing.
In response to the acronyms and abbreviations, the researchers have change the way to present the acronyms and abbreviations in each paragraph and the tables in the paper. The new acronyms and abbreviations has been presented in the table below.
|
Old |
The word |
New |
|
HUM |
Human performance |
HP |
|
EPE |
Economic performance |
EP |
|
KEY |
Key performance outcome |
KPO |
|
GOV |
Governance performance |
GP |
|
SER |
Service delivery |
SD |
|
SCS |
Services, Customers and Stakeholders |
SCS |
|
IAL |
Innovation and Learning |
IL |
|
SES |
Service user/stakeholder |
SUS |
|
WOR |
Workforce |
WORK |
|
EAE |
Environmental and Energy Importance |
EEI |
|
LEAD |
Leadership |
LEAD |
|
COW |
Community Welfare |
CW |
|
PPRE |
People, partnerships and resources |
PPR |
|
ENP |
Environment performance |
ENP |
|
CON |
Contribution to Education |
CE |
|
FINA |
Financial |
FINA |
|
INTE |
integration of corporate social responsibility into public sector scorecard |
ICP |
|
COR |
Corporate social responsibility disclosure |
CSRD |
|
ORG |
Organizational performance |
OP |
In term of the long grammatical structures, the researchers have change several sentences in the paper, such as line 16, 17, 21, 22, 23, 24, 27-29, 131, 132, 148, 178, 179, 183-184, 200, 232, 243, 252-255, 351, 586. Furthermore, in light of the adjustment in the methodology design, the limitation has been modified (line 706-710).
Round 3
Reviewer 1 Report
Dear author(s)
I appreciate the great lengths you have gone to in order to explain the variables and items and to restructure the introduction and background. Getting the acronyms and abbreviations just right can be a tedious undertaking, but one that I think is critically important. Now that I fully understand the context of the paper (and have hopefully assisted in making that understanding easier for others), I do not have any substantive comments to add.